



# 1 Inter-annual variation of aerosol pollution in East Asia and

## 2 its relation with strong/weak East Asian winter monsoon

Min Xie [1*], Lei Shu [1], Ti-jian Wang [1*], Da Gao [1], Shu Li [1], Bing-liang Zhuang [1], Anning Huang [1],
Dexian Fang [2], Yong Han [1], Mengmeng Li [1], Pu-long Chen [1], Zhi-jun Liu [1], Zheng Wu [2], Hua Lu [2]
[1] School of Atmospheric Sciences, CMA-NJU Joint Laboratory for Climate Prediction Studies,
Jiangsu Collaborative Innovation Center for Climate Change, Nanjing University, Nanjing 210023,
China
[2] Chongqing Institute of Meteorology and Science, Chongqing 401147, China

9  -------------------------------------------------------------------

[*] Corresponding authors. School of Atmospheric Sciences, Nanjing University, Nanjing 210023,
China. minxie@nju.edu.cn (M. Xie), tjwang@nju.edu.cn (T. J. Wang)
**Abstract:** Aerosol has become one of the major air pollutants in East Asia, and its spatial
distribution can be affected by the East Asian monsoon circulation. By means of the observational
analysis and the numerical simulation, the inter-annual variation of wintertime aerosol pollution in
East Asia and its association with strong/weak East Asian winter monsoon (EAWM) are
investigated in this study. Firstly, the Moderate Resolution Imaging Spectroradiometer/Aerosol
Optical Depth (MODIS/AOD) records during 2000-2013 are analyzed to reveal the inter-annual
variation characteristics of aerosols. It is found that there is an increasing trend of AOD in East
Asia over the last decade, implying the increasing aerosol loading in this region. The areas with
obvious increasing AOD cover the Sichuan Basin (SCB), the North China Plain, and most of the
Middle-Lower Yangtze River Plain in China. Secondly, the EAWM index (EAWMI) based on the
characteristic of circulation are calculated to investigate the inter-annual variations of EAWM. The
National Centers for Environmental Prediction (NCEP) reanalysis data are used in EAWMI
calculation and meteorological analysis. Nine strong and thirteen weak EAWM years are
identified from 1979 to 2014. In these strong EAWM years, the sea-land pressure contrast
increases, the East Asian trough strengthens, and the northerly wind gets anomalous over East
Asia. More cold air masses are forced to move southward by strengthened wind field and make
cool. In the weak EAWM years, however, the situation is totally on the opposite. Finally, the





effects of strong/weak EAWM on the distribution of aerosols in East Asia are discussed. It is
found that the northerly wind strengthens (weakens) and transports more (less) aerosols southward
in strong (weak) EAWM years, resulting in higher (lower) AOD in the north and lower (higher)
AOD in the south. The long-term weakening trend of EAWM may potentially increase the aerosol
loading. Apart from the changes in aerosol emissions, the weakening of EAWM should be another
cause that results in the increase of AOD over the Yangtze River Delta (YRD) region, the
Beijing-Tianjin-Hebei (BTH) region and SCB but the decrease of AOD over the Pearl River Delta
(PRD) region. Using the Regional Climate-Chemistry coupled Model System (RegCCMS), we
further prove that the intensity of EAWM has great impacts on the spatial distribution of aerosols.
In strong (weak) EAWM years, there is a negative (positive) anomaly in the air column content of
aerosol, with a reduction (increment) of -80 (25) mg·m$^{-2}$. The change pattern of aerosol
concentrations in lower troposphere is different from that at 500 hPa, which is related with the
different change pattern of meteorological fields in EAWM circulation at different altitude. More
obvious changes occur in lower atmosphere, the change pattern of aerosol column content in
different EAWM years is mainly decided by the change of aerosols in lower troposphere.
**Key words**:East Asian winter monsoon;Monsoon index;Aerosol;AOD;RegCCMS

**1. Introduction**

Atmospheric aerosol refers to the particulate matter in solid or liquid phase suspended in the

atmosphere with a diameter between 0.001-100 μm. It is not only a significant atmospheric
pollutant (Zhang et al., 2012b; 2013; Ding et al., 2013; 2016; Zhao et al., 2013; Guo et al., 2014;
Quan et al., 2014; Zheng et al., 2015), but also an important climate forcing factor that can directly
or indirectly affect the earth climate by influencing atmospheric radiation (Twomey, 1977;
Ramanathan et al., 2001a; Nakajima et al., 2003; Li, et al., 2007) , air temperature (Albrecht, 1989;
Giorgi et al., 2003; Liu et al., 2016), cloud physics (Fan et al., 2012; 2013; Nair et al., 2012),
precipitation (Rosenfeld, 2000; Rosenfeld et al., 2008; Giorgi et al., 2003; Qian et al., 2009;
Konwar et al., 2012), wind (Jacobson and Kaufman, 2006; Bollasina et al., 2011; 2014; ; Yang et
al., 2013), and atmospheric circulation (Allen et al., 2014; Niu et al., 2010; Song et al., 2014) etc.
On the other hand, changes of meteorological conditions (temperature, precipitation, and monsoon
circulation etc.) also can influence the emission, transport, chemical reaction and deposition



processes of aerosols, and thereby worsen the air quality (Jacob and Winner, 2009; Isaksen et al.,
2009; von Schneidemesser, 2015; Wu et al., 2016; Xie et al., 2017). For the above-mentioned
reasons, the relationship between aerosol pollution and climate system is acquired worldwide
attention in the scientific community (Isaksen et al., 2009; von Schneidemesser, 2015; Wu et al.,
2016; Li et al., 2016c). In the past decade, the interactions between aerosol and monsoon climate
has become the hot topic (Li et al., 2016c), especially in South Asia (Ramanathan et al., 2001a;
2001b; Ganguly et al., 2012; Nair et al., 2012; Manoj et al., 2012; Bollasina et al., 2011; 2014) and
East Asia (Nakajima et al., 2003; Lau et al., 2006; Li et al., 2007; 2009; 2016a; 2016b; Niu et al.,
2010; Zhang et al., 2010; 2012b; 2013; 2014; Zhao et al., 2010; 2013; Yan et al., 2011; Zhu et al.,
2012; Mu and Zhang, 2014; Song et al., 2014; Chen and Wang, 2015; Wang et al., 2015; Wu et al.,

2016).

East Asia is one of the most densely populous regions, and the homeland of one-third of the
world population (Li et al., 2011). In the past decades, the rapid development of economy,
industry and agriculture with expanding population in East Asia has resulted in large amounts of
anthropogenic aerosol emissions in this region (Li et al., 2016c). It was reported that the aerosol
concentration in East Asia (especially eastern China) is second to that of the cities in South Asia,
and the anthropogenic components (sulfate, nitrate and organics, etc.) account for a large
proportion of total aerosols (Zhang et al., 2008; 2012b; 2013). This high level of aerosol pollution
can exert much influence on regional atmospheric environment (Ding et al., 2013; 2016; Xie et al.,
2016; Zhu et al., 2017), weather (Ding et al., 2013; 2016) and climate (Nakajima et al., 2003; Lau
et al., 2006; Zhuang et al., 2013a; 2013b; Song et al., 2014; Wang et al., 2015; Li et al., 2007;
2009; 2011; 2016b). On the other hand, East Asia experiences the most remarkable monsoon
climate. The variation in monsoon circulation can not only directly affect the climatic
characteristics (air temperature, precipitation, and atmospheric circulation etc.), but also affect the
horizontal and vertical transport of atmospheric matters, such as moisture (Zhang, 2001; Fu et al.,
2006), cloud droplet (Tang et al., 2014), and air pollutants (Liu et al., 2003; Randel et al., 2010;
Bian et al., 2011) etc. Thus, the production, emission, transport and deposition processes of
aerosols can be significantly impacted by the East Asian monsoon circulation (Niu et al., 2010;
Zhang et al., 2010; 2013; 2014; Zhao et al., 2010; 2013; Yan et al., 2011; Zhu et al., 2012; Mu and
Zhang, 2014; Chen and Wang, 2015; Li et al., 2016a; 2016b; 2016c; Wu et al., 2016).



There have been lots of studies concerning the interactions between aerosol and monsoon
climate over East Asia. Some considered the mechanisms of the aerosol impact on monsoon
climate (Nakajima et al., 2003; Lau et al., 2006; Li et al., 2011; 2016b; Manoj et al., 2012; Song et
al., 2014; Wang et al., 2015). Some tried to reveal the effects of monsoon climate on aerosols (Niu
et al., 2010; Zhang et al., 2010; 2013; 2014; Zhao et al., 2010; 2013; Liu et al., 2011; Yan et al.,
2011; Zhu et al., 2012; Chen and Wang, 2015; Wang et al., 2015; Li et al., 2016a). However,
many of the previous studies about the later topic mainly focused on the impacts of summer
monsoon climate (Zhang et al., 2010; Zhao et al., 2010; Liu et al., 2011; Yan et al., 2011; Zhu et
al., 2012; Wang et al., 2015; Li et al., 2016c; Wu et al., 2016). In East Asia, high aerosol pollution
episodes usually occur in winter. Thus, how the East Asian winter monsoon (EAWM) circulation
modulates aerosols is worth to be investigated, and can help us comprehensively understand the
formation of aerosol pollution over East Asia in recent years.
Some researchers have gained improved knowledge of the effect of EAWM on aerosol
pollution (Mu and Zhang, 2014). For example, Zhao et al. (2013) and Zhang et al. (2013) pointed
out that the high concentration of local aerosols in North China can weaken the incoming
solar radiation on the ground, increase the atmospheric stratification stability, and in turn cause the
continuously and cumulatively increase of aerosols. Besides, the outward transport of aerosols is
weakened by the weak monsoon circulation in the winter, which also helps to cause the
continuous fog and haze weather in China. Zhang et al. (2014) analyzed the meteorological
conditions during the severe fog-haze periods over eastern China in January 2013. They concluded
that with weak winter monsoon circulation, the upper westerly jet slows down, vertical shear in
horizontal winds recedes, and thereby the development of synoptic disturbances and the vertical
mixing of the air masses are weakened. These anomalies in meteorology are all favorable for the
maintenance and the development of fog-haze over eastern China. Meanwhile, the anomaly of
south wind in the lower and middle level of troposphere hinders the outward transport of aerosols
as well. From these studies, it was found that the aerosol pollution episodes are inextricably linked
with the weak monsoon circulation, but the conclusion was just on basis of the individual aerosol
pollution episodes.
Several researchers have tried to understand the effect of EAWM on aerosol pollution in East
Asia by exploring the long-term variation trends of air pollutants and climate (Niu et al., 2010;





Chen and Wang, 2015; Li et al., 2016a). Based on the records of thirty years, Niu et al. (2010)
found that the frequencies of wintertime fog-haze events have doubled across eastern-central
China, while the speed of surface wind and the frequency of cold wave respectively decreased by
19% and 29% for the same period. They pointed out that weakening of the EAWM is likely a
major cause for the changes in meteorology, and has potential impact on the enhancing aerosol
loading and wintertime fog in China (Niu et al., 2010). However, they did not emphasize the
inter-annual variation of EAWM and aerosol, and could not reveal the exact different effects of
strong and weak EAWM on aerosols. Chen and Wang (2015) investigated haze days in North
China as well as the associated atmospheric circulations during 1960–2012, and mentioned that
the weakened northerly winds, the inversion anomalies in the lower troposphere, the weakened
East Asian trough in the midtroposphere, and the northward East Asian jet in the high troposphere
are the main causes leading to the winter haze. But, they studied the variation and the driving
factors in all seasons only based on the observation data of visibility. Special attention should be
paid to winter, and model simulation should be applied to probe the exact mechanism of the
EAWM impact on aerosols. Li et al. (2016a) investigated the inter-annual variation of wintertime
fog–haze events over eastern-central China from 1972 to 2014 and its association with EAWM.
They revealed that the stronger (weaker) the EAWM is, the less (more) the fog–haze events occur.
This phenomenon is related with the changes of near-surface winds, vertical shear in horizontal
winds, and divergence or convergence in the upper troposphere in different EAWM years (Li et al.,
2016a). However, this work was only based on the observational analysis of meteorological data,
and did not exactly present how EAWM impacts the distribution of aerosols. To better reveal the
mechanisms of the EAWM impact on aerosol, the inter-annual variation of EAWM, as well as the
difference in aerosol distribution in different EAWM years, should be discussed, and integrated
approach based on long-term observations and improved models is needed to further analyze the
mechanism (Li et al., 2016c).
The main purpose of this study is to improve our understanding of the effects of circulation
variation of EAWM on the distribution and transport of aerosols. By means of the observational
analysis of the Moderate Resolution Imaging Spectroradiometer/Aerosol Optical Depth
(MODIS/AOD) records and the National Centers for Environmental Prediction (NCEP) reanalysis
data during 2000-2013, as well as the Regional Climate-Chemistry coupled Model System





(RegCCMS) numerical simulation, we focus on (1) the long-term variation trend of aerosols in the
wintertime of East Asia, (2) the inter-annual variation of EAWM by identifying the strong/weak
EAWM years based on a EAWM index (EAWMI), and (3) the effects of strong/weak EAWM on
the distribution of aerosols. In this paper, detailed descriptions about the observational records for
aerosol and meteorology, the method to calculate EAWMI, and the adopted model with
configuration are illustrated in Section 2. The main findings, including the inter-annual variations
of AOD and EAWM, as well as the effect of EAWM on the distribution of aerosols in the winter
of East Asia, are given in Section 3. In the end, a brief summary is presented in Section 4.

**2. Data and methods**
**2.1 Aerosol optical depth records and meteorological data**
The MODIS/AOD monthly records from 2000 to 2013 are used to analysis the distribution
characteristic of aerosols in the winter of East Asia, and its inter-annual variation in the past
decade. The data can be collected from MODIS Collection 5.1 dataset, with wave band of 550 nm
and a horizontal resolution of $1° \times 1°$. Much work has been done to validate the feasibility of
MODIS aerosol products, and it was found that MODIS/AOD has reached the designed accuracy
with an error within $\pm0.05 \sim \pm0.20$ τ (Chu et al., 2002). The application of MODIS products in
China presents great territorial and seasonal differences, but the applicability of the products
reaches above 80% in the area with homogeneous surface and high-covered vegetation, which can
meet the error standard (>70%) of NASA (Wang et al., 2007). In general, the satisfying accuracy,
the high spatial resolution, and the high temporal coverage of MODIS/AOD make it widely
applied in scientific research for the regional distribution of aerosols (Tao et al., 2013; Li and Han,

2016).

In this study, the meteorological data are used to calculate the East Asian winter monsoon
index (EAWMI) that is used to identify the intensity of EAWM, and analyze the changes of
meteorological factors in the strong/weak EAWM years. The data are obtained from the NECP
global monthly reanalysis data from 2000 to 2013, including the large-scale meteorological
variables such as geopotential height, air temperature, zonal wind, meridional wind, sea level
pressure and precipitation etc., with a horizontal resolution of $2.5° \times 2.5°$ and a vertical resolution
of 17 levels. December, as well as January and February in the following year, is regarded as the



winter months.
**2.2 East Asian winter monsoon index**
EAWM is a complex atmospheric circulation system, the strength of which is affected by
diverse factors. The appropriate monsoon index is of great significance to explore the variation of
EAWM. There are lots of indices measuring the intensity of EAWM applied in previous studies.
They are mainly classified into 5 classes, which are on basis of the characteristic of circulation, the
characteristic of wind field, the characteristic of high pressure, the characteristic of sea-land
pressure contrast, and the integrated characteristic of winter monsoon system, respectively (Shao
and Li, 2012). According to their different research focuses of EAWM, these indices differ from
each other in the definition of monsoon intensity. The previous comparison showed that EAWMI
calculated by the characteristic of circulation or wind field can identify the strong/weak EAWM
better than other monsoon indices, and thereby these two kinds of indices (especially the first one)
have been widely applied in relevant studies (Shao and Li, 2012). Consequently, we choose the
EAWM index based on the characteristic of circulation to describe the anomaly of EAWM in this
work.
The EAWM originates from the periodic southward movement of the strong northeast air
stream in the front of the Siberian High. Thus, in terms of the circulation situation, the intensity of
EAWM can be manifested as the variations of the intensity of the East Asian trough at 500 hPa
and the Mongolia high near the surface. In this case, the activity of the East Asian trough at 500
hPa can be used to represent the state of EAWM (Yan et al., 2004; Wang and He, 2012). Based on
the work of Yan et al. (2004), the monthly averaged geopotential heights at 500 hPa in December,
January and Febuary is firstly standardized. Then the average value covering the area of
(30°–45°N, 125°–145°E) is calculated as follows:
$$\overline{H} = H_{500}(25 - 40^o\,N, 110 - 130^o\,N) \tag{1}$$
where, $\overline{H}$ is the average of geopotential heights at 500 hPa ($H_{500}$). Finally, $\overline{H}$ is standardized to
be the value between -1 and 1, as the following algorithm:
$$EAWMI = \frac{(\overline{H}_i - \mu)}{\sigma} \tag{2}$$
where, $\overline{H}_i$, μ and σ are the value to be standardized, the mean value and the standard deviation of





all sample points, respectively. EAWMI is the EAWM index used in this study. Previous
researches have proved that it can well demonstrate the characteristics of circulation and air
temperature in East Asia (Yan et al., 2004; Shao and Li, 2012).

**211  2.3 Regional climate chemistry modeling system and its simulation configuration**

The regional climate chemistry modeling system (RegCCMS) used in this study is an on-line
coupled model system composed of Regional Climate Model (RegCM3) with Tropospheric
Atmospheric Chemistry Model (TACM) (Li et al., 2009; Zhuang et al., 2013a; 2013b; Wang et al.,
2010; 2015). RegCM3 was developed by International Center for Theoretical Physics Research
Center (ICTP) in Trieste, Italy (Pal et al., 2007). Its dynamical core is based on the hydrostatic
version of the fifth generation Pennsylvania State University–National Center for Atmospheric
Research (PSU–NCAR) mesoscale model MM5. It adopts the terrain-following sigma coordinate,
and its radiation scheme takes the effects of greenhouse gases, aerosols, and ice clouds into
consideration (Giorgi et al., 2003; 2004a; 2004b). TACM includes complicated atmospheric
chemical and physical processes to deal with the emissions, transports, transformations and
depositions of trace gases and aerosols (Zhuang et al., 2013a; 2013b), and can be applied to
simulate the effects of primary pollutant emissions (eg., $SO_2$, $NO_x$ and VOCs etc.) on the regional
pollution of gases, aerosols, and acid deposition (Wang et al., 2010; 2015). RegCM3 and TACM
are two-way coupled in RegCCMS. RegCM3 provides meteorological data to drive TACM,
including air temperature and solar radiation data used in the calculation of chemical reaction rates,
cloud cover and actinic flux data used in the calculation of photolysis rate, moisture data needed in
some atmospheric chemistry reactions, wind field and turbulent field required to deal with the
advection, diffusion and dry deposition processes of pollutants, and cloud and rainfall parameters
required to deal with liquid phase chemical and wet scavenging processes. On the other hand,
TACM outputs the spatial and vertical distributions of trace gases and aerosols, some of which
have special influence on atmospheric radiation transfer and can affect the radiation process in
RegCM3. Numerous previous studies have shown that RegCCMS have a satisfying performance
in the simulations of climate change and air quality. It can better simulate the regional
meteorological fields, the concentrations of air pollutants, and the climatic effects of various
aerosols, including black carbon, nitrate, sulfate, and primary organic carbon (Li et al., 2009;
Wang et al., 2010; 2015; Zhuang et al., 2013a; 2013b).



RegCCMS is applied in this study to simulate the differences in meteorological field and
aerosol distribution between strong and weak EAWM years, and further reveal the effects of
EAWM on the transport and the distribution of aerosols over East Asia. Figure 1 shows the grid
setting of the simulated domain, which covers most of East Asia, with the center point at 34.5°N,
116.8°E, horizontal grids of 121 × 90, and grid spacing of 50 km. From the surface to the model
top (50 hPa), there are 18 vertical sigma layers, with the σ values of 1.0, 0.99, 0.98, 0.96, 0.93,
0.89, 0.84, 0.78, 0.71, 0.63, 0.55, 0.47, 0.39, 0.31, 0.23, 0.16, 0.1, 0.05 and 0.0.

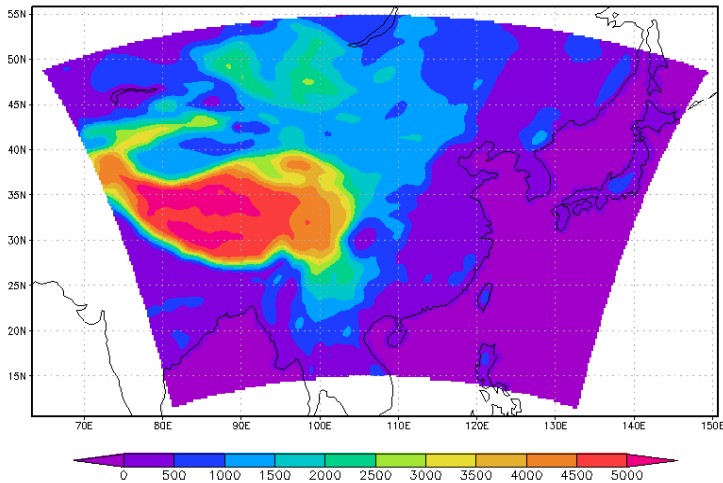


**Figure 1. The grid setting of the simulated domain in RegCCMS.**

The major selected physical options in RegCM3 include the ACM2 boundary layer scheme
(Holtslag and Boville, 1990), the CCM3 radiation scheme (Kiehl et al., 1996), the BATS
land-surface scheme (Dickinson, 1993), the Grell cumulus parameterization scheme (Grell and
Devenyi, 2002). The initial and boundary conditions of meteorological fields are obtained from
NCEP global reanalysis data with $2.5^o \times 2.5^o$ resolution.
For TACM, the finite positive definite difference method of Smolarkiewicz for the advective
term (Smolarkiewicz, 1984), the Crank-Nicolson scheme for the vertical diffusion term, and the
central difference scheme for the horizontal diffusion term (Press et al., 1992) are used. As for the
chemical options, a condensed gas-phase chemistry scheme, a simple aqueous chemistry scheme,
and the aerosol model ISORROPIA are adopted. The gas-phase chemistry scheme includes 20



species and 36 reactions (Wang et al., 2010). The aqueous chemistry scheme considers the soluble
gases absorbed by cloud and rain droplets as well as the aqueous oxidation of $SO_2$ and $NO_x$ (Wang
et al., 2010). ISORROPIA is a thermodynamic equilibrium model that can simulate sulfate and
nitrate aerosols (Nenes et al., 1998). The resistance analogy method named the big leaf model
(Walmsley and Wesely, 1996) is used to simulate dry deposition velocities. The in-cloud and
below-cloud scavenging of aerosols are calculated as a function of rainfall amount (Wang et al.,
2010). More details of the schemes can be found in the previous studies (Li et al., 2009; Wang et
al., 2010; 2015; Zhuang et al., 2013a; 2013b). The emission inventory used in this study is based
on the work of Zhuang et al. (2013a; 2013b) and Wang et al. (2015). It is basically obtained from
the inventory that is developed for the NASA INTEX-B mission (Zhang et al., 2009), and includes
the emissions of aerosols and associated precursors over China in 2006 with the monthly
variations of pollutants.

**3. Influence of EAWM on the distribution of aerosols based on observational analysis**

**3.1 Characteristic of the inter-annual variation of wintertime aerosols in East Asia**

Figure 2 shows the time series of average wintertime AOD in East Asia from 2000 to 2013.
From the linear trend (red dotted line in Figure 2), it is clear that the pollution level of aerosols in
East Asia gets significantly increased in the past ten years, which should be mainly caused by the
increased emissions of aerosols associated with the rapid development of economy over East Asia
in these years (Zhang et al., 2012b). In addition, some previous studies also revealed that the
increasingly aerosol loading may be tightly related to the weakening of EAWM during the same
period (Niu et al., 2010; Li et al., 2016a). Figure 2 shows the obvious inter-annual variation of the
wintertime AOD from 2000 to 2013 as well, with the maximum mean value of 0.44 in 2007 and
the minimum value of 0.36 in 2001. This suggests that the anomalous monsoon circulation may
play great roles in the inter-annual variation of aerosol loading in this region, which is further
discussed in Section 3.4 in detail.




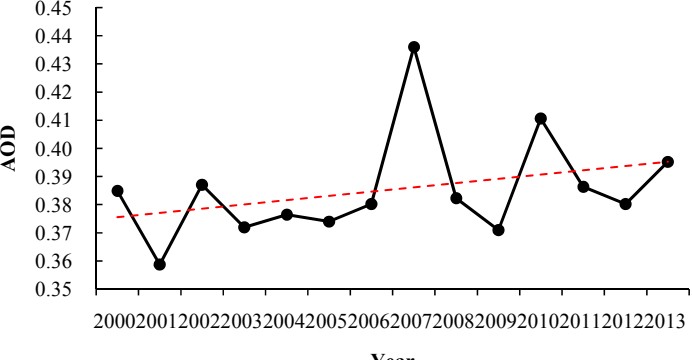


**Figure 2. Inter-annual variation (black solid line) and linear trend (red dotted line) of AOD in the winter of**
**East Asia from 2000 to 2013.**


The spatial distributions of the average value (Figure 3a) and the changes (Figure 3b) of
wintertime AOD over East Asia during the period of 2000-2013 are demonstrated in Figure 3. As
shown in Figure 3a, the spatial distribution of AOD shows a clear regional feature over East Asia.
Although the values of AOD differ in different months, the areas with high AOD value are mainly
concentrated in the Sichuan Basin (SCB), around Bohai Bay, and in the Middle-Lower reaches of
Yangtze River in China. Meantime, from Figure 3b, it is found that the wintertime AOD over East
Asia shows a long-term rising trend, with a significantly increase in North China, Central China,
SCB and the Yangtze River Delta (YRD) region. To sum up, the areas with heavy aerosol loading
in East Asia mainly consist of the Beijing-Tianjin-Hebei (BTH) region (115-120$^{\circ}$E, 35-41$^{\circ}$N),
YRD (117-122°E, 30-34$^{\circ}$N), and SCB (103-107°E, 28-32$^{\circ}$N). Besides, the Pearl River Delta (PRD)
region (111-116$^{\circ}$E, 18-24$^{\circ}$N) is a remarkable developed urban agglomeration in South China, and
its aerosol pollution can represent the level of fog-haze pollution in the south of East Asia.


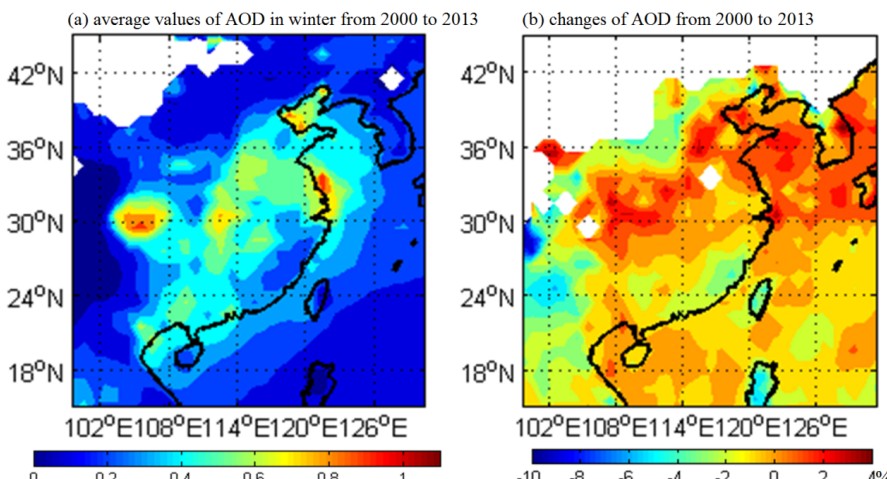


**Figure 3. The spatial distribution of the average value and the changes of AOD over East Asia during the**
**period of winter 2000-2013.**

Figure 4 presents the inter-annual variation of average wintertime AOD in the above four
typical regions of East Asia from 2000 to 2013. The following Table 1 gives the corresponding
statistics. It can be seen that the value of AOD in YRD is much higher than that in other regions,
with the average, maximum and minimum value being 0.55, 0.59 (in 2000) and 0.48 (in 2004),
respectively. As for other three regions, the respective average, maximum and minimum AOD
values are 0.44, 0.51 (in 2012) and 0.31 (in 2003) for SCB, 0.42, 0.51 (in 2012) and 0.32 (in 2003)
for BTH, and 0.36, 0.43 (in 2003) and 0.28 (in 2012) for PRD. Except for PRD, the AOD values
in other three regions show a rising trend. As shown in Figure 4 and Table 1, the largest ten-year
increment is 13.1% in BTH, followed by 9.4% in SCB and 2.4% in YRD. Based on the satellite
remote sensing data, it was found that there exists the largest increase of air pollutant
concentrations in BTH and YRD during the last ten years, which has caused the increase of
particulate matter concentrations in these two regions (Zhang et al., 2012a). Increased emissions
produced by human activities have resulted in the increase of AOD and haze weather in SCB, and
the special topography of basin can also lead to the aggravation of fog-haze pollution in this
region to a certain extent because of the constraint effect of topography on the transport and
diffusion of aerosols (Chen et al., 2014). As for the PRD region, there is a slight reduction in AOD
during recent years with the reduction of -3.6%, which may be mainly attributed to the scientific





and effective control of air quality in this region for a long time.

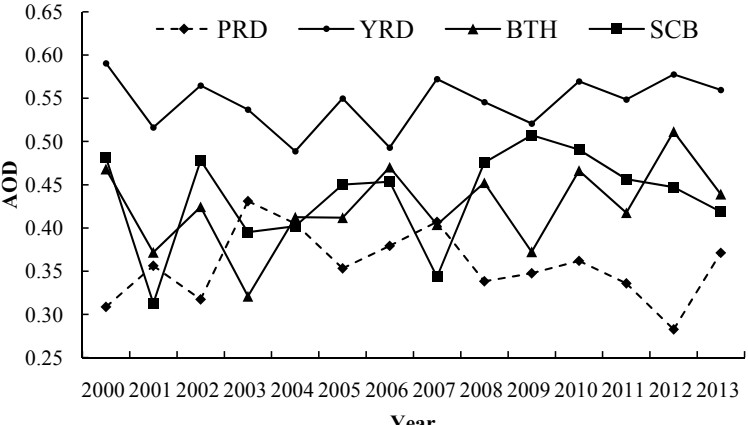


**Figure 4. Average value of wintertime AOD in the four typical regions from 2000 to 2013.**


**Table 1. Statistics values of wintertime AOD in the four typical regions during the period of 2000-2013.**

|  | Maximum | Minimum | Average | 10-year increment |
|---|---|---|---|---|
| BTH | 0.51 | 0.32 | 0.42 | 13.1% |
| YRD | 0.59 | 0.49 | 0.55 | 2.4% |
| SCB | 0.51 | 0.31 | 0.44 | 9.4% |
| PRD | 0.43 | 0.28 | 0.36 | -3.6% |


**3.2 Inter-annual variation of EAWM**

Figure 5 shows the inter-annual variation of EAWMI from 1979 to 2014. According to the

definition of EAWMI in Section 2.2, when EAWMI is larger than 0, the positive anomaly of 500
hPa geopotential height occurs over East Asia (25~40°N, 110~130°E). In this case, the East Asian
trough is shallow, and its upper northwest air stream is comparatively weak. Thus, there is a weak
winter monsoon circulation in East Asia, which tends to result in weak cold air activities. On the
contrary, when EAWMI is lower than 0, there is a deeper East Asian trough that is related with a
stronger upper northwest air stream. The stronger cold air activities frequently take place, and
thereby result in the stronger winter monsoon. Thus, the value of EAWMI in a year can be used as
the criterion to distinguish if the year is a weak or strong winter monsoon year.

From the linear variation depicted in Figure 5, the value of EAWMI shows an increasing

trend, with the value larger than 0.5 and even above 1 in recent years. The trend means that the



East Asian winter monsoon circulation has been significantly weakened during recent decades.
Moreover, the comparison of Figure 5 and Figure 3 further implies that the weakened winter
monsoon may increase the wintertime AOD loading over East Asia, which is in agreement with
the findings of Xu et al. (2006), Niu et al. (2010) and Li et al. (2016a).

As shown in Figure 5, the EAWMI also presents a strong inter-annual variation, with the

maximum value of 1.5, the minimum value of -2.2 and the maximum inter-annual difference of
3.7. According to the definition of Wang and Chen (2014), the year with the value of
EAWMI >0.5 (< -0.5) is identified to be the weak (strong) EAWM year. Consequently, there are 9
strong EAWM years (1979, 1980, 1982-1985, 1996, 2001 and 2010) and 13 weak EAWM years
(1986-1989, 1997, 1998, 2002, 2005-2006, 2008 and 2012-2014) being identified from 1979 to
2014. Obviously, there are more strong EAWM years before 1986, with the lowest value of
EAWMI being -2.2 in 1985. After 1986, EAWM tends to be weaker, and there are more weak
EAWM years instead. This conclusion corresponds to the previous findings (Nakamura et al.,
2002; Jhun and Lee, 2004; Wang et al., 2009).

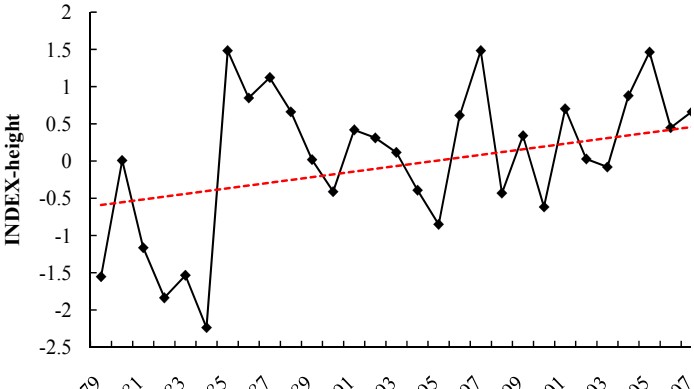


**Figure 5. Inter-annual variation of EAWMI from 1979 to 2014.**

**3.3 Difference in meteorological fields between strong and weak EAWM years**

The EAWMI used in this study can well illustrate the variation of circulation resulting from

the EAWM anomaly. Figure 6 presents the spatial distribution of correlation coefficient between
EAWMI and some meteorological factors over East Asia. As shown in the figure, there exists a
distinct positive correlation between EAWMI and 500 hPa geopotential height (Figure 6a), surface
temperature (Figure 6a), precipitation (Figure 6c), and sea level pressure (Figure 6d) in most area





of (20~40°N, 108~135°E), with the largest correlation coefficient being 0.8, 0.8, 0.6 and 0.8,
respectively.

For the correlation between EAWMI and 500 hPa geopotential height, it seems that the

increase in the value of EAWMI corresponds to the shallow 500 hPa East Asian trough, the less
cold air activity, and the weak EAWM circulation. Under this circumstance, the surface air
temperature gets increased (Figure 6b), implying that the surface temperature in weak EAWM
years is higher than that in strong EAWM years. As mentioned in other researches, this EAWMI
indeed can reflect the anomaly of average winter temperature over East Asia to some extent (Yan
et al., 2004; Shao and Li, 2012). With respect to the correlation between EAWMI and sea level
pressure, it is found that the thermal low over the west Pacific Ocean strengthens as the increased
EAWMI (Figure 6c). Generally, there are a cold high on the land and a thermal low on the sea in
the winter of East Asia. Thus, the positive correlation in Figure 6c also means that the sea-land
pressure contrast decreases in weak EAWM years while it increases in strong EAWM years. When
it comes to precipitation (Figure 6d), it increases when the value of EAWMI rises up. It can be
concluded that there is more rainfall in weak EAWM years than in normal years over East Asian
continent. On the contrary, the East Asian continent features a dry cold climate with low surface
temperature in strong EAWM years. Overall, it is convinced that the EAWMI can reflect the
variation anomaly of 500 hPa geopotential height, surface temperature, sea-level pressure and
precipitation over East Asian continent to some extent.





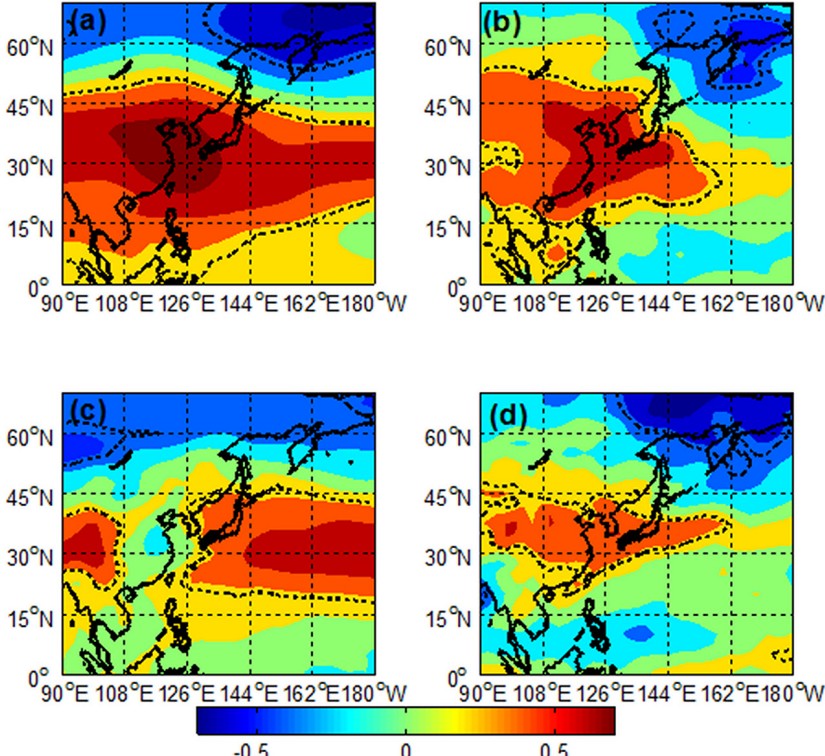


**Figure 6. The correlation coefficient between EAWMI and meteorological factors, including (a) 500 hPa**
**geopotential height, (b) surface temperature, (c) sea-level pressure and (d) precipitation.**

The differences in meteorological factors between strong and weak EAWM years are
examined, targeted at the 9 strong and the 13 weak EAWM years mentioned in Section 3.2. Figure
7 shows the average anomaly of the meteorological factors in the strong and the weak EAWM
years. Apparently, the spatial distributions of circulation and meteorological factors in the strong
EAWM years are almost completely opposite to those in the weak EAWM years. Figure 7a and b
demonstrate the anomaly of 500 hPa geopotential height field in different EAWM years. In the
strong EAWM years (Figure 7a), there are a negative anomaly in the East Asian trough and a
positive anomaly in high latitude area of East Asia, which can be in favor of the deepening and
strengthening of the East Asian trough. However, in the weak EAWM years (Figure 7b), there is a
positive anomaly in the core area of the East Asian trough, making the trough shallower and
weaker. Figure 7c and d present the wind field anomaly at the 850 hPa level in different EAWM
years. It seems that the north wind anomaly prevails at 850 hPa over East Asia, and the northwest



and the north wind dominate over the east of China in the strong EAWM years (Figure 7c).
Meanwhile, there also exists a west wind anomaly near 20°N and an east wind anomaly near 50°N
in North Pacific Ocean. But as shown in Figure 7d, the 850 hPa wind field in the weak EAWM
years is contrary to that in the strong EAWM years. These findings are in agreement with the
results obtained from Yan (2004). For the anomaly of the sea-level pressure field, Figure 7e
illustrates that there is a negative anomaly of sea-level pressure in the strong EAWM years. The
sea-level pressure is lower in these years than in normal years, which means that there are larger
sea-land pressure contrast and stronger winter monsoon circulation in strong EAWM years.
However, Figure 7f presents the different pattern in the weak EAWM years, that is, the sea-land
pressure contrast is smaller and the winter monsoon circulation is generally weaker. Figure 7g and
h provide the surface air temperature anomaly. It is clearly observed that there is a negative
anomaly of the average winter temperature in the mainland of China in the strong EAWM years
(Figure 7g). The drop of air temperature should be related with the fact that more cold air masses
may be transported from north to south (Figure 7c). In the weak EAWM years, the opposite
conditions appear (Figure 7h).
On the whole, the anomaly of atmospheric circulation in the strong EAWM years can be
characterized as: (1) there is a positive anomaly in the Siberian High, a negative anomaly in
sea-level pressure over Pacific Ocean area, and an apparent increase of sea-land pressure contrast;
(2) there is an obvious anomaly of cyclonic circulation in 850 hPa wind field over East Asia, and
thereby the northerly wind prevails over East China; (3) the 500 hPa geopotential height gets
decreased over East Asia and increased over the Pacific Ocean area, which synthetically lead to
the strengthening of the East Asian trough; (4) there is stronger north wind in northeast and north
China, which allows more cold air mass to invade northward and results in a sharp fall in air
temperature in most of East Asia. The anomalies of the meteorological factors in the weak EAWM
years are almost completely opposite to the above characteristics.





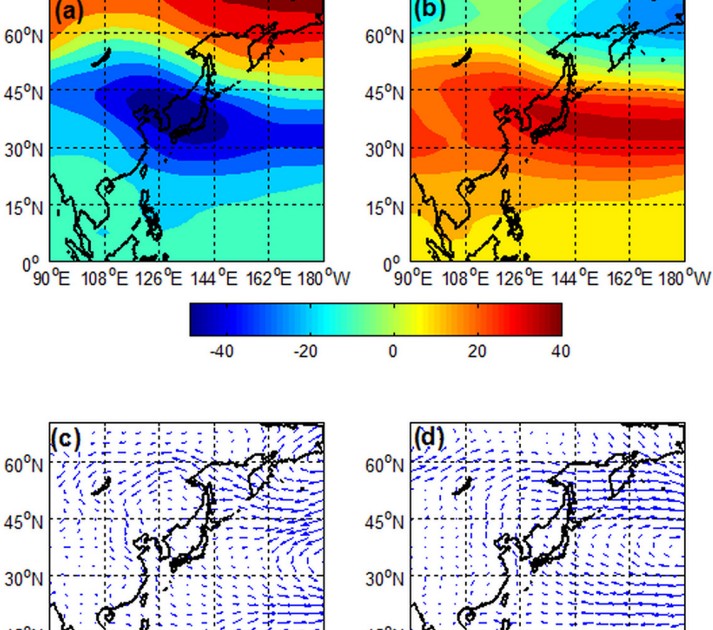




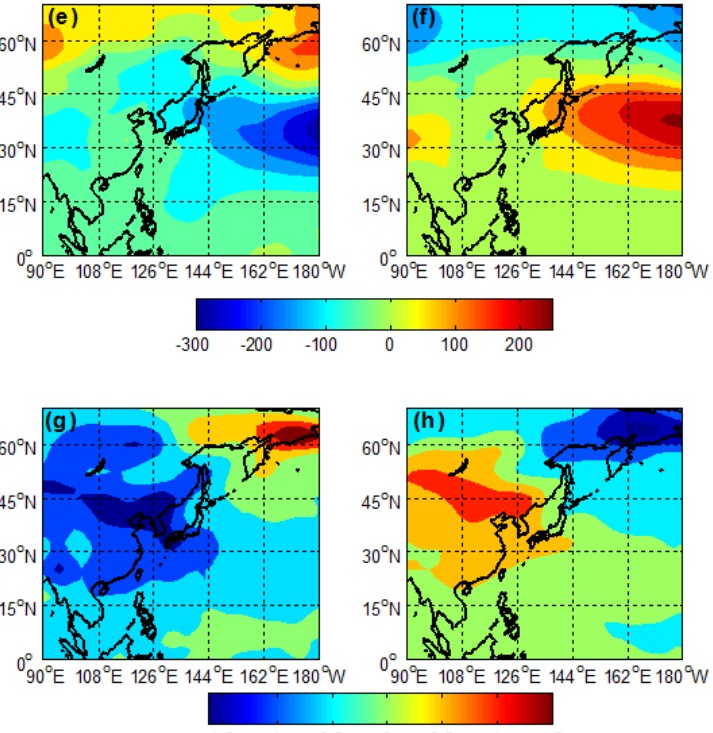


**Figure 7. Anomaly of meteorological factors in strong EAWM years (a, c, e and g) and weak EAWM years (b,**
**d, f and h), including (a-b) 500 hPa geopotential height (unit: m), (c-d) 850hPa wind field, (e-f ) sea-level**
**pressure (unit: Pa) and (g-h) surface air temperature (unit: °C).**

**3.4 Difference of aerosol distribution between strong and weak EAWM Years**
The spatial distribution of aerosols is not only highly connected to the local anthropogenic
emissions, but also to the long-range transport influenced by atmospheric circulation and the
scavenging effects of rainfall. Given that East Asia is located in the famous monsoon climate
region, we analyze the difference of the distribution of aerosols between strong and weak EAWM
years, aimed to figure out the effects of EAWM on aerosol pollution in this region. Only the
MODIS/AOD data after 2000 are available in this study, so the data in 2 strong EAWM years
(2001 and 2010) and 4 weak EAWM years (2002, 2005, 2006 and 2008) are used in this section to
conduct composite analysis.
Figure 8a and b display the wintertime average distribution of AOD over East Asia in the
above-mentioned strong and weak EAWM years. Admittedly, the regions with high AOD in the
strong and the weak EAWM years are generally unchanged, and mainly concentrated in the



well-developed areas of East China, such as those around Bohai Bay, the North China Plain, and
the Middle-Lower reaches of Yangtze River. It indicates that the anthropogenic emissions are
mainly responsible for the high values of AOD instead of the winter monsoon circulation. Figure
8c shows the difference of AOD distribution between the strong EAWM years and the normal
years (that is anomaly), while Figure 8d illustrates that between the weak EAWM years and the
normal years. In strong EAWM years, the aerosol loading is lower in the northern area of East
Asia and slightly higher in the southern area than that of normal years (Figure 8c). Thereinto, there
are obvious negative anomalies in North China, SCB and the middle reach of Yangtze River,
implying that aerosols get decreased in these areas. But in the weak EAWM years, there are
positive anomalies in these three regions, which may be attributed to the increase of aerosol
concentrations. Meanwhile, there are negative anomalies in most of southern China, suggesting
that the aerosol loading decreases and is lower than that of normal years.
Figure 8e further displays the differences of aerosol distribution between the strong and the
weak EAWM years, by means of subtracting from AOD in the weak EAWM years from that in the
strong EAWM years. The difference distinctly reveals that there are fewer aerosols over North
China, SCB, and the middle reach of Yangtze River while more aerosols over the south of East
Asia in the strong EAWM years than in the weak EAWM years (Figure 8e). The difference can be
explained by the prevailing wind over East Asia. In the strong EAWM years, there is a prevailing
northerly wind, which can transport more aerosols in the north to the south area of East Asia. In
the weak EAWM years, however, the wind is not strong, and more pollutants may be trapped and
accumulated in the north because of the stagnant weather condition.
On account that the East Asian winter monsoon circulation tends to be weakened in the past
decades (as shown in Figure 5), the weak of EAWM should be another cause that results in the
increase of AOD over YRD, BTH and SCB but the decrease of AOD over PRD during this period
(as shown in Figure 4). As discussed in Section 3.3, the weakening of EAWM circulation is highly
related with the weakening of the East Asian trough and the Siberian High, the reduction of the
sea-land pressure contrast over Pacific Ocean area, and the decrease of the northerly wind over
East China. Thus, the weather tends to be more stagnant in recent years, and thereby more aerosols
remain and lead to higher AOD values in the source areas (such as YRD, BTH and SCB). In
winter, more pollutants are emitted from the surface in the north because of more heating demands,
so the aerosol pollution in the north is generally worse than that in the south. The decrease of the
northerly wind results in the decease of transport of aerosols from the north to the south, which
may contribute to the decrease of the AOD value in PRD.

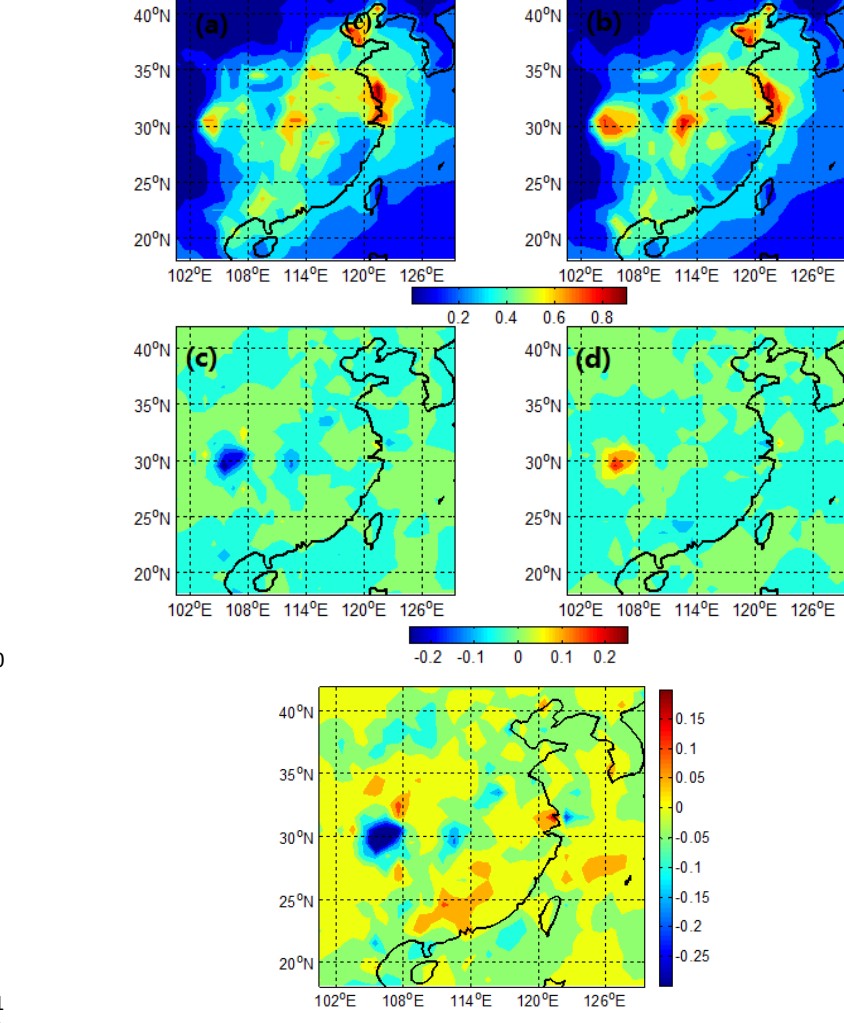



**Figure 8. Composite analysis of AOD, including average AOD distribution in the strong (a) and the weak (b)**
**EAWM years, the anomaly of the distribution of AOD in the strong (c) and the weak (d) EAWM years, and**
**(e) difference of AOD distribution between the strong and the weak EAWM years. Here, the strong EAWM**
**years include 2001 and 2010, while the weak EAWM years include 2002, 2005, 2006 and 2008.**

**4. Simulation of the effects of EAWM on aerosol distribution**
To reveal the possible impacts of EAWM on the transport and the distribution of aerosols, the



regional climate chemical model RegCCMS is used to model the concentrations of aerosols in the
strong and the weak EAWM years. The simulations are conducted for every winter from 2001 to
2010. The emissions are assumed to remain fixed in different years to eliminate the influence of
emission changes. The model outputs are averaged to represent the mean distribution of aerosols
over the last 10 years, the strong EAWM years (2001 and 2010), and the weak EAWM years
(2002, 2005, 2006 and 2008). Because the emissions keep fixed, the differences of the distribution
of aerosols between the strong EAWM years and the weak EAWM years can be considered to be
only caused by the anomaly of the EAWM circulation. Other simulation settings are listed in
Section 2.3.
**4.1 Model validation**

In order to evaluate the model performance, the NCEP reanalysis data is adopted to verify the

accuracy and applicability of the modeling results from RegCCMS. Figure 9 shows the
comparisons between the reanalysis data and the model results in winter (December, January and
February) for the multi-year mean (from 2001 to 2010) values of surface air pressure, temperature
and wind field at 850 hPa, air temperature at 500 hPa. It is certain that the results from RegCCMS
simulations are consistent with those from the NCEP reanalysis data. The best performance can be
found in the simulations for the surface air pressure (Figure 9a). The simulated high and low
values of surface air pressure, as well as the spatial distribution, match well with the reanalysis
data. The simulated air temperature fields at 850 hPa and 500 hPa are also in agreement with the
NCEP data except for those in the Qinghai-Tibet Plateau (Figure 9b and c). For the wind field at
850 hPa, the model performs well in the simulation of wind direction, wind speed and wind field
structure in the wintertime of East Asia (Figure 9d). In a word, the modeling results of RegCCMS
show good correlation with the observations, suggesting that RegCCMS is able to capture and
reproduce the features of meteorological fields in different monsoon years.



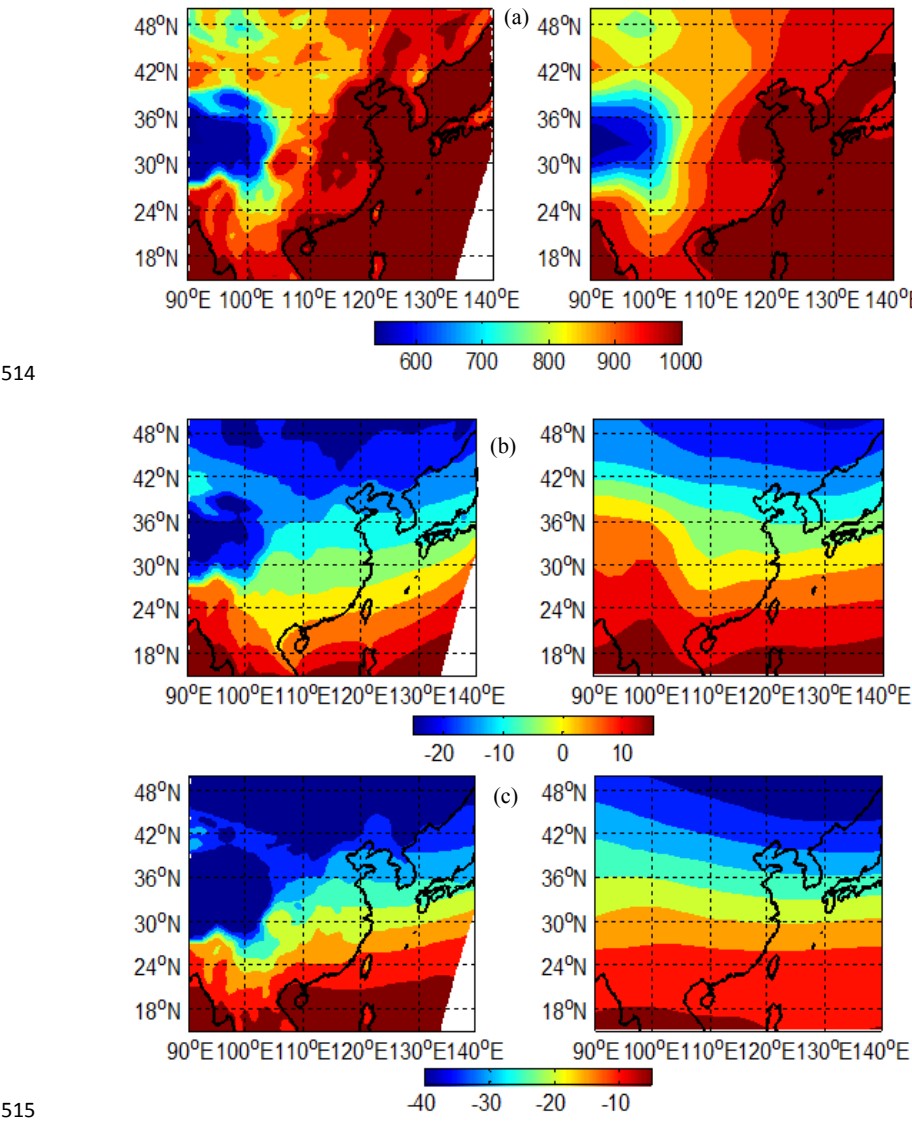







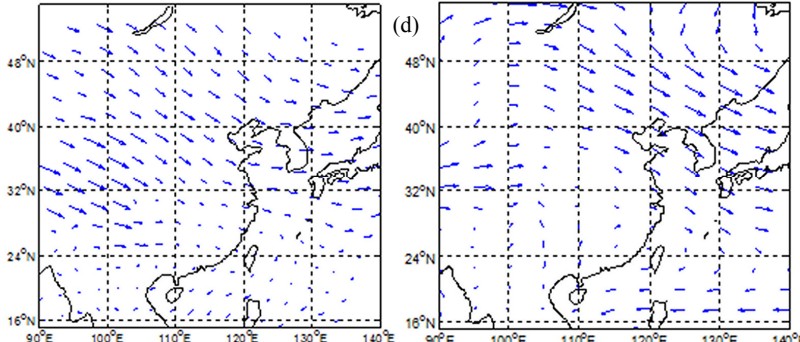


**Figure 9. Numerical simulation (left) and observation (right) of meteorological fields, including (a) surface**
**air pressure (unit: hPa); (b) 850 hPa temperature (unit: °C); (c) 500 hPa temperature (unit: °C) and (d) 850**
**hPa wind field (unit: m s$^{-1}$).**

Figure 10 provides the simulated differences in the distribution of meteorological factors

between the strong and the weak EAWM years, including air temperature at surface and wind field
at 850 hPa. For the wind field at 850 hPa, there is evident cyclonic circulation and northerly air
stream in East China. As a result, the existing northerly wind anomaly is conducive to transport
more cold air masses from the north to the south. Thus, there is a more obvious negative
temperature anomaly in the mainland of East Asia in the strong EAWM years than in the weak
EAWM years. The change areas of air temperature are in agreement with the changes of
atmospheric circulation, which further proves that much stronger northerly wind can result in the
southward invasion of cold air masses and the drop of air temperature in the strong EAWM years.
The above findings from the modeling results coincide with those from the observational data
analysis in Section 3.3, implying that RegCCMS performs well in the simulations for the anomaly
of meteorological fields in the strong and the weak EAWM years.





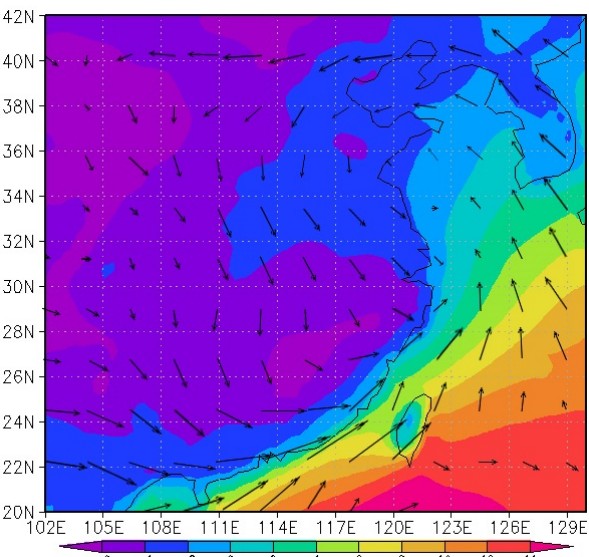


**Figure 10. Simulated differences in the distribution of meteorological factors between the strong and the weak EAWM years, including surface air temperature (unit: K) with 850 hPa wind field.**


Figure 11 exhibits the multi-year average aerosol column content in the strong and the weak

EAWM years, which is simulated by RegCCMS. The column content of aerosol is calculated from

surface to the model top. The simulated high values of aerosol loading occur in SCB, Central

China and the coastal areas of East China. Meanwhile, low aerosol loading can be found in the

coastal areas of the provinces of Fujian and Guangdong. The simulated distribution pattern is

generally consistent with that achieved from the satellite observation (Figure 3 and 8). However,

the modeling results do not well catch the maximum values of AOD observed around Bohai Bay

and the coastal areas of YRD. This bias may be attributed to the fact that AOD is not only

correlated with the concentration of aerosol but also affected by moisture. In all, even though the

modeling results and the observation AOD differ in units, it still can be found that RegCCMS well

explains the overall spatial distribution features of wintertime aerosol loading.





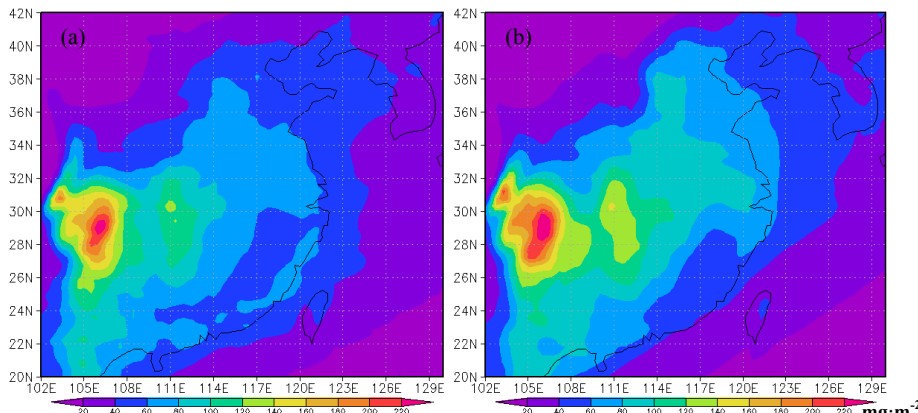

**Figure 11. Simulated multiyear mean aerosol column contents in (a) the strong EAWM years and (b) the weak EAWM years by RegCCMS. The column content of aerosol is calculated from surface to the model top.**

**4.2 Effects of strong and weak EAWM on the distribution of aerosols**

As shown in Figure 11, there is no obvious difference in the aerosol spatial distribution between the strong and the weak EAWM years. The highest aerosol loading generally occurs in SCB. However, there still exists some difference. Both the area coverage of high values and the intensity of aerosol loading are larger in the weak EAWM years, with aerosol column content reaching high value as 200 mg·m$^{-2}$ and covering the area of 104°E-107°E and 27°N-30°N (Figure 11b). For the strong EAWM years, the high values (higher than 200 mg·m$^{-2}$) are limited to the area of 106°E-107°E and 28°N-29°N (Figure 11a). The result is the same for the differences in regions with secondary high value. For the area of 110°E-113°E and 27°N-32°N, the aerosol column content ranges from 140 to 160 mg·m$^{-2}$ in the weak EAWM years, while the value is lower than 140 mg·m$^{-2}$ in the strong EAWM years.

Furthermore, Figure 12a and b demonstrate the anomaly of aerosol column content and 850 hPa wind field in the strong and the weak EAWM years, which greatly differ in the spatial distribution. As shown in Figure 12a, in the strong EAWM years, there is a negative anomaly in the area east to 110°E and north to 28°N, with the maximum reduction over -10 mg·m$^{-2}$. In contrast, there is a positive anomaly in the weak EAWM years for the same area, with the maximum increment over 20 mg·m$^{-2}$. In addition, in the strong (weak) years, there is a negative (positive) anomaly in the region of 26°N-30°N near SCB, with the maximum change value over



-25 (30) mg·m$^{-2}$. As to the wind anomaly at 850 hPa, it can be found that there are a northerly
wind anomaly and an increase in the component of north wind in the strong EAWM years, which
should be linked with the effects of cyclonic circulation over Ease Asia. On the contrary, East Asia
is influenced by the anticyclonic circulation in the weak EAWM years. There appears a south
wind anomaly and a weaker northerly wind than those in normal years.

Figure 12c shows the difference in the distribution of aerosol column content as well as the

wind at 850 hPa between the strong and the weak EAWM years. It appears that the northwest
wind in the area north to 28°N is stronger in the strong EAWM years than in the weak EAWM
years, which helps to transport more aerosols from the north to the south. In consequence, the
decrease of aerosol loading can be found in most land areas of East Asia, with the highest
decrement over -60 mg·m$^{-2}$ in North China and SCB in the strong EAWM years. Meanwhile, the
west wind in the area south to 28°N is strong as well, which can further transport aerosols to the
coastal areas in the south. Thus, the synthetic impacts of the north and the west wind cause the
significant increases of aerosols in the coastal areas of Fujian, Guangdong and Guangxi, with the
typical increment over 25 mg·m$^{-2}$. This driving effect of wind on the distribution of aerosols
results in the higher AOD value in the south and lower in the north in the strong EAWM years,
which has been displayed in Figure 8.

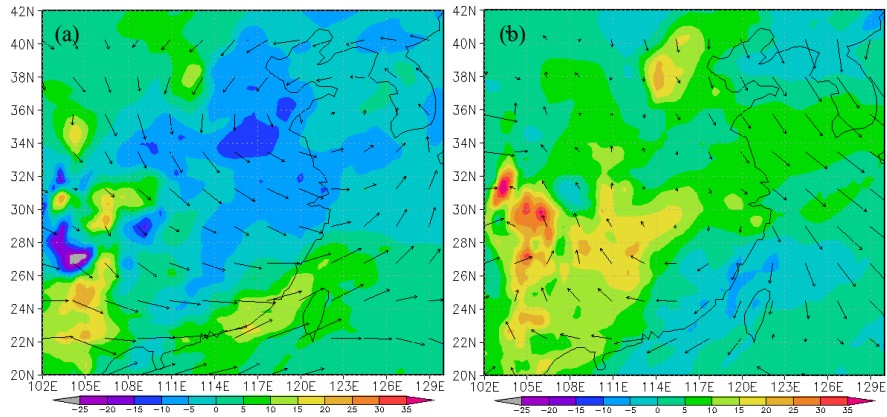


(c)





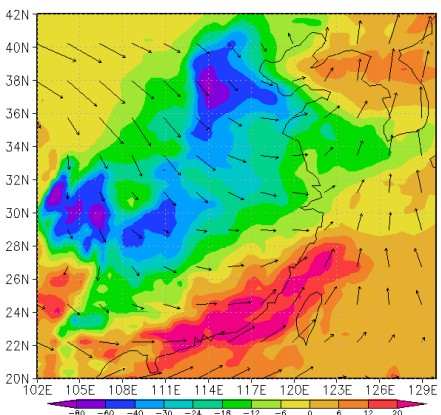


**Figure 12. Simulated aerosol column content in the strong and the weak EAWM years, including the aerosol column content anomaly and the 850hPa wind anomaly in (a) the strong and (b) the weak EAWM years, and (c) the difference in aerosol column content and 850 hPa wind between the strong and the weak EAWM years (aerosol column content of weak years subtracted from that of strong years).**


Figure 13 illustrates the aerosol concentration anomalies at surface, 850 hPa and 500 hPa in
the strong and the weak EAWM years, which are calculated by the aerosol concentrations at the
corresponding altitude subtracting from the multi-year average values. As shown in Figure13a and
b, there is an obvious negative anomaly at surface over East Asia in the strong EAWM years, with
the relatively high decreases in the North China Plain and the highest reduction over -23 $\mu g \cdot m^{-3}$ in
BTH. In the weak EAWM years, there is a positive anomaly in the east of East Asia instead, with
the maximum increment of 35 $\mu g \cdot m^{-3}$. As for the wind field anomaly at surface, there is a stronger
northerly wind over East Asian continent in the strong EAWM years than in normal years
(Figure13a). This wind anomaly in the strong EAWM years may help to reduce the aerosol
column content by carrying more aerosols from the inland to the sea areas in the southeast of East
Asia. Worthy of note is that there is a relatively larger positive anomaly covering the area of
104°E-106°E, 26°N-28°N, which should be related to the east wind anomaly in this region.
However, in the weak EAWM years, the surface wind slows down, hindering the outward
transport of aerosols and resulting in much more accumulation of aerosols on the land (Figure
13b).
As for the changes at 850 hPa, there is a negative anomaly of aerosol concentration in the
North China Plain and the reaches of Yangtze River while a positive anomaly in the areas north to
26°N in the strong EAWM years. As shown in Figure 13c, this change pattern of aerosol



distribution should be attributed to the positive anomaly of the northerly wind in most land areas
of East Asian continent. In the weak EAWM years, however, the mainland of East Asian continent
is affected by anticyclonic circulation (Figure 13d). Consequently, there appears a positive aerosol
concentration anomaly in Southwest China, Central China and the Middle-Lower reaches of
Yangtze River. Meanwhile, a negative anomaly occurs in the coastal areas of Fujian and
Guangdong provinces. The region with the biggest difference covers the area of 109°E-111°E,
29°N-32°N, with the decrement (increment) of -10 (23) $\mu g \cdot m^{-3}$ in the strong (weak) EAWM years.
Affected by the different changes of monsoon circulation at different altitude, the change
patterns of aerosol and wind at 500 hPa are different from those in lower troposphere. As shown in
Figure 13e, there are stronger northeast wind in the area north to 39 °N and stronger southwest
wind in the area south to 30 °N in the strong EAWM years. Thus, more aerosols accumulate in the
areas between 30°N and 39°N, and thereby there is a positive aerosol concentration anomaly. On
the contrary, there is stronger northwest wind in the mainland of East Asian continent in the weak
EAWM years, which results in a negative aerosol concentration anomaly north to 30°N and a
positive anomaly south to30°N.
To sum up, in the lower troposphere, there is enhanced horizontal wind in the strong EAWM
years, which transports more aerosols to the southeast coastal areas and reduces aerosol
concentrations on the land. However, the aerosols cannot be transported outward in the weak
EAWM years and accumulate around the source areas, increasing the aerosol concentrations in the
mainland of East Asia than those in normal years. The change pattern of aerosol concentrations is
different at 500hPa, which is related with the different change pattern of meteorological fields
affected by the upper part of EAWM circulation. The bigger difference in aerosol concentrations
between the strong and the weak EAWM years occurs in lower troposphere. The changes of
aerosols range from -14 to 30 $\mu g \cdot m^{-3}$, -10 to 23 $\mu g \cdot m^{-3}$, and -0.06 to 0.14 $\mu g \cdot m^{-3}$ at surface, 850
hPa, and 500 hPa, respectively. Thus, the change pattern of AOD (or simulated aerosol column
content) in different EAWM years is mainly decided by the change of aerosols in lower
troposphere.








**Figure 13. The simulated anomaly of wintertime aerosol concentration and wind field in the strong EAWM years (a, c and e) and the weak EAWM years (b, d and f) at surface (a and b), 850hPa(c and d), and 500hPa (e and f) (unit: μg·m$^{-3}$).**


**5. Conclusion**



This paper investigates the impacts of EAWM on the distribution of wintertime aerosol in

East Asia on the basis of observational data analysis and numerical simulations. MODIS/AOD is

used to analyze the spatial distribution and long-term variation trends of aerosols over East Asia.

The EAWM index identified by the characteristics of circulation is adopted to study the long-term

variation of EAWM. The different characteristics of meteorological fields in the strong and the

weak EAWM years are analyzed by using the NCEP reanalysis data. Combined the results from

observations and RegCCMS simulations, the differences in distribution anomaly for aerosols

between strong and weak EAWM years, and the potential transport effects of monsoon circulation

are discussed. The main conclusions are as follows.

(1) There exists an increase trend in wintertime AOD over East Asia, which shows obvious

inter-annual variation characteristics with the maximum value of 0.44 in 2007 and the minimum

value of 0.36 in 2001. In winter, high AOD values mainly occur over SCB, the North China Plain

and most of the Middle-Lower reaches of Yangtze River. Moreover, there are obvious increases of

AOD in these regions.

(2) With the aid of the EAWM index, it can be summarized that there are 9 strong EAWM

666        years (1979, 1980, 1982-1985, 1996, 2001 and 2010) and 13 weak EAWM years (1986-1989,

1997, 1998, 2002, 2005-2006, 2008, and 2012-2014) during the period from 1979 to 2014. The

intensity of winter monsoon is stronger before 1986 and gets weakened since 1986. The

meteorological conditions differ in different EAWM years. In the strong EAWM years, the

sea-land pressure contrast gets increased, the East Asian trough gets strengthened, and the

northerly wind anomaly dominates over East Asia. The stronger wind transports more cold air

masses southward and causes the air temperature drop in the mainland of East Asia. The change

patterns of meteorological factors are just the opposite of those in the weak EAWM years.

(3) Though higher aerosol loading in winter is largely ascribed to the huge emission

generated by human activities, the EAWM circulation can change the distribution of aerosols as

well. The northerly wind speeds up over East Asia in the strong EAWM years and transports

aerosols southward, resulting in AOD higher in the south and lower in the north of East Asia. In

contrast, in the weak EAWM years, the northerly wind slows down and allows more aerosols to

accumulate in the North China Plain, resulting in AOD higher in the north and lower in the south.

The long-term weakening trend of EAWM may potentially increase the aerosol loading over YRD,





BTH and SCB, while causes the decrease of AOD over PRD.
(4) It is further confirmed by numerical simulation that the stronger (weaker) northerly wind
transports more (less) aerosols southward and there appears a negative (positive) aerosol column
content anomaly in mainland China in the strong (weak) EAWM years. The difference in aerosol
column content between the strong and the weak EAWM years ranges from -80 mg·m$^{-2}$ to
25mg·m$^{-2}$. The change pattern of aerosol concentrations in lower troposphere is different from that
at 500 hPa, which is related with the different change pattern of meteorological fields in EAWM
circulation at different altitude. The changes of aerosols range from -14 to 30 μg·m$^{-3}$, -10 to
23μg·m$^{-3}$ and -0.06 to 0.14μg·m$^{-3}$ at surface, 850 hPa and 500 hPa, respectively. The change
pattern of aerosol column content in different EAWM years is mainly decided by the change of
aerosols in lower troposphere.
It has been proved that the variations of EAWM can directly affect the transport, diffusion,
deposition and chemical reaction processes of aerosols. This paper is only concerned about the
effects of strong and weak EAWM on the transport of aerosols. Therefore, future researches
considering the effects of EAWM on other processes of aerosols are needed to deepen the
discussion. Moreover, the data for aerosols in various types with high resolution have been
available, thus more specific studies about effects of EAWM on different kinds of aerosols should
be strengthened in the future as well.

**Acknowledgments**
This work was supported by the National Natural Science Foundation of China (41475122,
91544230, 41621005), the National Key Research and Development Program of China
(2016YFA0602104), the open research fund of Chongqing Meteorological Bureau (KFJJ-201607),
and the Fundamental Research Funds for the Central Universities. The authors would like to thank
the anonymous reviewers for their constructive and precious comments on this manuscript.

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
