# Peer review of "Inter-annual variation of aerosol pollution in East Asia and"

_Atmospheric Chemistry and Physics, 2017_

## Referee Comment (RC1) · Anonymous Referee #1 · 21 Jul 2017

Presentation quality is poor. I am surprised the manuscript can pass the quality control process with such poor figure quality.

1. Figure 5 is a critical figure to determine the weak/strong monsoon years, but not a complete one. The printed version only shows data from 79-05, but caption shows it should cove till 2014. Without a complete figure, it is difficult to evaluate many parts of manuscript. 2. Figure 3 is also an important figure, not clear what "changes of AOD" mean? 14 years trend or 2013 minus 2000? 3. Figure 1, what shade shows? Terrain? 4. Figure 2, averaged over which region? 5. Figure 4, how did you define the 4 regions (PRD, YRD, BTH, SCB)?

[Figure]

Section 4, there are not any evaluation of model performance provided in simulating aerosol concentration, PM2.5, AOD and precipitation?

Scientific significance is poor. Finding is not new and many studies have been done in this area but not being discussed or compared. Idea and analysis in this manuscript apparently follow these two paper, but not being cited.

Jeong, J.I. and Park, R.J. (2017). Winter monsoon variability and its impact on aerosol concentrations in East Asia. Environ. Pollut. 221: 285–292.

Mao, Y.-H., Liao, H., and Chen, H.-S.: Impacts of East Asian summer and winter monsoons on interannual variations of mass concentrations and direct radiative forcing of black carbon over eastern China, Atmos. Chem. Phys., 17, 4799-4816, https://doi.org/10.5194/acp-17-4799-2017, 2017.

---

## Referee Comment (RC2) · Anonymous Referee #2 · 12 Sep 2017

The deterioration of air quality (caused by aerosols) and the change in monsoon climate are two environmental threats to the people living in East Asian monsoon region. This paper presents an interesting and important study on the inter-annual variation of EAWN and its impact on the aerosol pollution in East Asian combining observation and numerical simulation. The outcome could lead to a better understanding of the interaction between aerosol and monsoon in particular East Asia. The manuscript fits the aim and the scope of ACP. Before being published, it still needs revision. The comments are listed as below.

Introduction -Very good review on the impact of EWAM on aerosol transport and fog

formation based on both long-term observation and air pollution episode, but lacks a critical discussion of the mechanism how EWAM will affect aerosol pollution and Vice Versa. -The language needs some editing from a native speaker.

Methodology -The data used for EAWMI calculation and meteorological analysis are from 1979 to 2014, but the used AOD data are from 2000 to 2013. The time length is not the same. Why? -Could you explain in more detail how the two-way coupling is achieved? A model validation should also be presented to demonstrate the robustness of the simulation.

Results and discussion -Line282- please clarify why the anomalous monsoon circulation may play a role in the inter-annual AOD? -Figure 3b- what do you mean the change of AOD over 2000-2013? could you show BTH, SCB,YRD and PRD in the figure 3? -Figure 4- as the AOD is only collected between 2000-2013, could you compare AOD and EAWMI within this time scale? Same results could be obtained? -I have some concerns on directly linking AOD with EAWMI as there are many influential factors determining AOD especially emissions. Key question is how to eliminate the influence of other factors to derive a more conclusive conclusion.

Language The English should be polished. There are many grammar errors. Some are listed as follows.

Line 23, "the inter-annual variations of EAWM" is better to be "the inter-annual variation of EAWM". Line 32-33, "resulting in higher (lower) AOD in the north and lower (higher) AOD in the south" is better to be changed as "resulting in higher (lower) value of AOD in the north and lower (higher) in the south". Line 58, "changes of" is better to be "changes in". Line 60 "the air quality", "the" is better to be deleted. Line 109, "periods" is better to be "period". Line 146, please keep "distribution" and delete "and transport". Line 175, "NECP" should be "NCEP". Line 342, "above 1" is better to be revised as "over 1". Line 381, "a dry cold climate" is better to be revised as "a dry and cold climate". Line 406, "obtained" should be deleted. Line 459, the first from should be deleted. Line

[Figure]

499, "the NCEP reanalysis data is adopted to" should be "the NCEP reanalysis data are adopted to".

---

## Referee Comment (RC3) · Anonymous Referee #3 · 20 Sep 2017

In this paper, the authors examined the interannual variability of aerosols over East Asia, using MODIS AOD data, and NCEP reanalysis, in relationship to the strength of the East Asian Winter Monsoon (EAWM). They concluded that a recent long-term weakening of the EAWM contributed to increased aerosol in the central and south-western China, Yangtze River Delta, Beijing-Tianjin-Hebei (BTH) and Szechuan Basin (SCB), and decreased aerosol in southern China. They also conducted numerical experiments using RegCCMS model to simulate strong and weak EAWM from 2000-2013, under conditions of prescribed aerosol emission for all years, to corroborate with their observational results. However, I don't see much similarity between the observed (Fig. 8) and the simulated (Fig. 12) AOD. Overall, this study does not add much new insight,

but more confusion to previous studies on similar topics. I recommend a rejection of the paper in its current form. However, given that the subject matter is important, and the combined approach of observation analysis and modeling is valid, I encourage a re-submission, with major revision and additional analysis, along the following lines:

1. The use 500 hPA geopotential height variation over a small region in the subtropics an mid-latitudes to define EAWM index in Eq (1) maybe problematic. This index tends to give too much weight to the mid-latitude, and less to the tropical influence on the EAWM. In this paper, the authors emphasized the increased (decreased) in low-level meridional winds during strong (weak) monsoon over East Asia, in affecting the aerosol transport and distribution. The changes in meridional winds between strong and weak EAWM as defined by the authors EAWM index, is mainly confined to the eastern part of mdlatitude East Asia, with little signal, such as cold air outbreak during strong EAWM, over tropical East Asia. The EAWM is a large-scale phenomenon, covering much larger area than the domain used in the present analysis. A better index should include the magnitude of the slp of the Siberian High , contrasting with the low slp over the North Pacific, and the Maritime Continent (see for example, Wang and Chen 2013, and others). Such an EAWM index will give a much sharper contrast between the anomalous northerly/southerly flow over the entire East Asia region. Using different indices is also likely to change the years of strong and weak EAWM years used in their subsequent analysis. The authors need to consider more carefully the proper choice of the EAWM index used for their analysis, including evaluating how the use of different EAWM indices may affect their results.

2. The short-term MODIS AOD data record (2000-2013) and therefore limited samples of strong (2 years), and weak (4 years) of EAWM are not likely to yield robust results. The only strong signal is found over the Szechuan Basin. This may be due to one year of very strong signal of one sign over the region, dominating the sample. What is the level of statistical confidence of the observed AOD differences in different regions? Statistical confidence should also be computed for the modeling results.

[Figure]

3. The authors discussed aerosol transport as the only factor (besides emission) that can affect AOD distribution. A strong vs. weak EAWM is likely to be associated with changes in atmospheric conditions such as stability, relative humidity, rainfall and aerosol residence time, etc., in affecting AOD. New insight may be gained by examining changes in some of these factors in affecting AOD between strong and weak EAWM.

4. The paper needs better organization and more careful English editing. Section 2.3 on RegCCM should be put in Section 4, before the discussion of the model results. The word " prove" is improperly used in many placed and should be avoided, since the results do not really prove anything. At best they "show" or "suggest" certain line of reasoning. Please make sure all figures are probably labeled. In some, e.g. Fig. 8, 9 , 12, the labels are missing or obscured, and the wind vectors , e.g. Fig. 10, are masked by the dark coloring.